# Simulation of observed climate changes in 1850-2014 with climate model INM-CM5

Evgeny Volodin[1*], Andrey Gritsun[1]

[1] Institute for Numerical Mathematics, (INM RAS, Gubkina 8, Moscow 119333, Russia)

* Correspondence to EvgenyVolodin, Institute for Numerical Mathematics, (INM RAS, Gubkina 8, Moscow 119333, Russia, tel: +7-495-9383904, fax: +7-495-9381821, e-mail: volodinev@gmail.com)

**Abstract**

Climate changes observed in 1850-2014 are modeled and studied on the basis of seven historical runs with the climate model INM-CM5 under the scenario proposed for Coupled Model Intercomparison Project, Phase 6 (CMIP6). In all runs global mean surface temperature rises by 0.8 K at the end of the experiment (2014) in agreement with the observations. Periods of fast warming in 1920-1940 and 1980-2000 as well as its slowdown in 1950-1975 and 2000-2014 are correctly reproduced by the ensemble mean. The notable change here with respect to the CMIP5 results is correct reproduction of the slowdown of global warming in 2000-2014 that we attribute to change in ocean heat uptake and more accurate description of the total Solar irradiance in CMIP6 protocol. The model is able to reproduce correct behavior of global mean temperature in 1980-2014 despite incorrect phases of the Atlantic Multidecadal Oscillation and Pacific Decadal Oscillation indices in the majority of experiments. The Arctic sea ice loss in recent decades is reasonably close to the observations in just one model run; the model underestimates Arctic sea ice loss by the factor 2.5. The spatial pattern of the model mean surface temperature trend during the last 30 years looks close to the one for the ERA Interim reanalysis. The model correctly estimates the magnitude of stratospheric cooling.

Keywords: climate, model, simulation, temperature, climate indices, sea ice extent

### 1. Introduction

Noticeable climate changes were observed during the last century. The main feature of these changes is global warming and it is widely accepted that its most probable reason is increase of anthropogenic greenhouse gas concentration (Bindoff et al. 2013). The nature of

several other important changes is not as clear and is still under discussion. Global warming was not uniform in time, there are two well known periods of acceleration: 1920-1940 and 1980-2000, and two periods with stabilization of global mean temperature: 1950-1975 and 2000-2014. The reason of this oscillatory behavior is still debated. In Wilcox et al. (2013) it is shown that period of climate stabilization in 1950-1975 can be connected with the increase of anthropogenic $SO_2$ emission in Europe and North America as well as with stratospheric volcanic eruptions (Bindoff et al. 2013), while decrease of warming in 2000-2014 could be attributed to slowdown of methane and tropospheric ozone concentration increase rate. On the other hand the ensemble of CMIP5 model runs (with the all mentioned aspects of aerosol and greenhouse gas forcing taken into account) continues to rise global temperature in 2000-2014 albeit with the slower rate (Checa-Garcia et al. (2016)).

Another point of view on this problem is that the acceleration and deceleration of global warming could be a manifestation of internal climate variability with the time scale of 60-70 years (Meehl et al. 2011). The Atlantic Multidecadal Oscillation (AMO) and Pacific Decadal Oscillation (PDO) are the most known drivers of internal variability in the climate system on multidecadal time scales. Indeed, Dong and McPhaden (2017) showed the importance of AMO-like and PDO-like internal variability for local temperature in the North Atlantic and the North Pacific but questioned its ability to produce a significant anomaly in global mean temperature. A connected question is to what extent the observed long-term variability of the AMO and PDO patterns is an internal process, or is forced by some external factors. There is some evidence (Ting et al. 2014) that negative values of the AMO index in 1950-1970 could be attributed to the enhanced $SO_2$ emission in Europe and North America.

One of the most intriguing features of recent climate changes is a rapid decrease of Arctic sea ice area since year 2000 coupled with strong Arctic warming. Similar Arctic warming was also observed in the middle of the 20th century. The ensemble of CMIP5 models underestimates the amount of sea ice loss in 2000s by a factor of two (Bindoff et al. 2013, Stroeve et al. 2012). The INMCM4 climate model (Volodin et al. 2013) that participated in CMIP5 also strongly underestimates Arctic sea ice extent loss in the beginning of the 21st century. On the other hand the INMCM4 (and other CMIP5 models, Stroeve et al. 2012) demonstrates loss of Arctic sea ice of comparable magnitude at different times (in the middle of the 20th century for INMCM4). Moreover, similar sea ice loss event was produced by INMCM4 during a preindustrial control run. So, the question is to what extent the 21st century Arctic sea ice degradation is due to the internal climate variability and whether the next generation of models with new CMIP6 forcing recommendations is able to reproduce sea ice changes in the beginning of 21st century?

Regional climate changes during the last several decades also show some interesting features. For example, in 2000-2014 there is almost no winter warming in the majority of Eurasia with respect to the previous decades and even a small cooling was observed in some places. One possible reason could be the response of atmospheric dynamics to Arctic sea ice loss (Overland et al. 2011). However this hypothesis is questioned by other studies (see McCuscer et al. (2016) as an example).

The aim of this study is to analyze basic features of climate changes during the 1850-2014. The data for the analysis (ensemble of seven historical runs) was produced by the new climate model INM-CM5 being an incremental upgrade of the INMCM4. We are mainly focusing on the question of how global mean surface temperature (GMST) changes are reproduced with the new forcing protocols proposed for CMIP6 and how these changes are connected with the reproduction of other features of the climate system mentioned above (i.e. AMO and PDO variability etc).

## 2.    Model and data.

The climate model INM-CM5 (Volodin et al. 2017A; Volodin et al. 2017B) was used in this study. In the atmosphere, it has spatial resolution of 2x1.5 degrees in longitude and latitude, and 73 levels in vertical, with the uppermost level at 0.2 hPa. In the oceanic block, spatial resolution is 0.5x0.25 degrees and 40 levels in vertical. The model includes an interactive aerosol block (Volodin and Kostrykin 2016), where concentrations of 10 aerosols are calculated. In numerical experiments discussed below only the first aerosol indirect effect (the influence of aerosol on cloud drop radius) is taken into consideration. Model description and analysis of simulation of the present day climate can be found in Volodin et al. (2017B).

Let us discuss now a climate change modeling experiment for years 1850-2014. Time series of $CO_2$, $CH_4$, $N_2O$, $O_3$, stratospheric volcanic sulfate aerosol concentration, total Solar irradiance (TSI) and solar spectrum, as well as anthropogenic emissions of $SO_2$, black carbon and organic carbon were prescribed as recommended for the historical run of CMIP6 (https://esgf-node.llnl.gov/search/input4mips/). Seven model runs were started with different initial conditions obtained from long preindustrial run, where all external forcings were prescribed at the level of year 1850. The length of preindustrial run was several hundred years, so the upper oceanic layer was adjusted to atmospheric model conditions, but it is not the case for the deep ocean. A small trend of model climate is visible because of deep ocean adjustment to upper oceanic and atmospheric conditions – a common situation for simulation of historical climate with present day climate models. The obvious reason for multiple integrations is to

separate the role of natural variability and external forcing in climate changes. When data of
seven model runs are consistent with each other, then one can expect that the phenomenon of
interest is a manifestation of (or response to) an external forcing. If there is a noticeable
difference between different model runs, then a role of natural variability is crucial. To estimate
statistical significance of near surface temperature trend, a t-test at 99% level was used. The
variance of 5 year means was calculated from 1200 years of preindustrial run.
Observational data of GMST for 1850-2014 used for verification of model results were
produced by HadCRUT4 (Morice et al. 2012). Monthly mean sea surface temperature (SST) data
ERSSTv4 (Huang et al. 2015) are used for comparison of the AMO and PDO indices with that of
the model. Data of Arctic sea ice extent for 1979-2014 derived from satellite observations are
taken from Comiso and Nishio (2008). The stratospheric temperature trend and geographical
distribution of near surface air temperature trend for 1979-2014 are calculated from ERA Interim
reanalysis data (Dee et al. 2011).
**3. Results.**
The most important measure of climate changes is the global mean surface temperature.
Observed GMST demonstrates the well known acceleration of warming in 1920-1940 and 1980-
2000 and small warming or even small cooling in 1945-1970 and 2000-2014. The ensemble of
CMIP5 models (Bindoff et al. 2013) shows less significant slowdown in warming in 2000-2014.
In particular, the INMCM4 model (Volodin et al. 2013) demonstrates gradual warming starting
from 1920.
With the new CMIP6 protocols all seven INM-CM5 model runs demonstrate fast
warming in 1980-2000 with the rate close to the observations and GMST stabilization in 2000-
2014 and 1950-1970 (Fig.1). The only significant difference in the new CMIP6 forcings at the
beginning of 21$^{st}$ century with respect to CMIP5 ones is the change in the TSI. Before year 2000
CMIP5 and CMIP6 TSI show almost identical behavior (CMIP5 Solar forcing can be found at
https://pcmdi.llnl.gov/mips/cmip5/forcing.html). In 2001-2008 TSI recommended for CMIP6 is
about 0.3 W/m$^2$ lower than the one for CMIP5 (Fig.2). For 2009-2014 CMIP5 scenario
suggested repetition of previous Solar cycle that gives the value of the TSI almost 1 W/m$^2$ above
the one recommended for CMIP6. An additional model run with anthropogenic aerosol
emissions fixed at the level of year 1850 shows gradual GMST rise in 1950-1970 together with
its stabilization 2000-2014 (not shown). The later fact supports the hypothesis that correct
reproduction of GMST changes in 2000-2014 is due to the corrected CMIP6 treatment of the
TSI. Another factor that stabilizes GMST in 2000-2014 in INM-CM5 is the heat flux to the
ocean (Fig.3) having the values of 0.3-0.7 W/m$^2$ (higher than in any period of 20th century). The
CMIP5 historical experiment with INMCM4 model shows gradual increase in the ocean heat
uptake during 1980-2005 rather than its abrupt jump in 1995-2005 seen in Fig.3. Note that Yan
et al. (2016) showed that according to the available observations slowdown in GMST increase in
1998-2013 can be explained by the increased ocean heat uptake that could be estimated as 0.7
W/m$^2$ for 1993-2010 according to Rhein et al. (2013).

7          Better representation of GMST stabilization in 1950-1970 (Fig.1) in simulations with

INM-CM5 with respect to the INMCM4 can be explained by incorporation of the new aerosol
block in the model that resulted in a more sophisticated treatment of anthropogenic and volcanic
aerosol interaction with atmospheric radiation. Fast warming in 1920-1940 similar to the
observed one can be seen in four model runs, while other three runs show warming earlier or
later. These results suggest that the observed acceleration of warming in 1920-1940 is probably
due to combination of external forcing and natural variability.

14         Keeping in mind the arguments that the GMST slowdown in the beginning of 21$^{st}$ century

could be due to the internal variability of the climate system let us look at the behavior of the
AMO and PDO climate indices. Here we calculated the AMO index in the usual way, as the SST
anomaly in Atlantic at latitudinal band 0N-60N minus anomaly of the GMST. The model and
observed 5 year mean AMO index time series are presented in Fig.4. The well known oscillation
with a period of 60-70 years can be clearly seen in the observations. Among the model runs, only
one (dashed purple line) shows oscillation with a period of about 70 years, but without
significant maximum near year 2000. In other model runs there is no distinct oscillation with a
period of 60-70 years but period of 20-40 years prevails. As a result none of seven model
trajectories reproduces behavior of observed AMO index after year 1950 (including its warm
phase at the turn of the 20$^{th}$ and 21$^{st}$ centuries). One can conclude that anthropogenic forcing is
unable to produce any significant impact on the AMO dynamics as its index averaged over 7
realization stays around zero within one sigma interval (0.08). Consequently, the AMO dynamics
is controlled by internal variability of the climate system and cannot be predicted in historic
experiments. On the other hand the model can correctly predict GMST changes in 1980-2014
having wrong phase of the AMO (blue, yellow, orange lines on Fig.1 and 4).

30         More coherent behavior of model trajectories after year 1980 could be seen for the North

Atlantic (45-65N) temperature (Fig.5). Indeed the temperature deviates from its 1850-1899 mean
by 1.5 sigma in the early 2000s. The NA temperature index in the model shows notable
oscillations with periods of about 30-40 and 60-80 years (close to the 25 and 80 years for the
observations), three trajectories have correct strongly positive NA temperature anomalies in the
21$^{st}$ century.

Another important climate feature that could be responsible for the changes of the GMST growth rate is the PDO measured by its index defined as normalized projection of the SST anomaly on a specific pattern in the North Pacific at 20-60N. The 5 year average PDO index for observations and model data is presented in Fig.6. For the observations, one can see maxima at years 1930-1940 and 1980-1995 and a prolonged minimum during 1950-1975. None of the model trajectories reflects observed time series of the PDO index by the same reasons discussed earlier in the paragraph devoted to the AMO. Again model does not need correct PDO index dynamics to predict GMST behavior.

One of the most intriguing observed features of ongoing climate changes is the fast summer Arctic sea ice extent decrease in the beginning of 21$^{st}$ century. The ensemble of CMIP5 models underestimates the rate of decrease of Arctic summer ice area by the factor of two. Model INMCM4 participated in CMIP5 also underestimates the value of Arctic sea ice extent decrease significantly (Volodin el al. 2013). In newly obtained INM-CM5 data (Fig.7) we see qualitatively the same behavior of the Arctic sea ice as the average rate of the sea ice loss is underestimated by the factor of the two to three.  Though, in one model run (purple) the magnitude of decrease is similar to the one in the observations (reduction from 7-7.5 million km$^2$ in 1980s to 4-5.5 million km$^2$ in 2000s). In other runs Arctic sea ice loss is underestimated by factor of 1.5-3, and in one run (green) one can even see some increase of Arctic sea ice area during last decades. Our results suggest that the rapid decrease of Arctic sea ice extent near year 2000 was partially induced by an external forcing however the role of internal variability can be very important (the range of the sea ice extent year-to-year variability could be estimated as 3.0 million km$^2$).

The stratosphere is more sensitive to global changes than the troposphere. One can see in the observations stratospheric cooling by several degrees during the last decades. In the ERA Interim reanalysis data (Fig.8) the global and annual mean temperature at 5 hPa in year 2014 is 3K lower than in year 1979. All model runs show gradual decrease of stratospheric temperature during all period of historical run from 1850 to 2014, but the rate of decrease in 1979-2014 is highest and equal 2.5K that is slightly below in absolute value than the observed one. This strong decrease is consistent in all model runs, and is likely produced by combined effects of $CO_2$ increase and ozone decrease. Oscillations of global mean temperature at 5 hPa with period of 10-12 years represent prescribed Solar cycle.

One of the characteristic features of climate changes in recent decades is a specific geographical pattern of surface temperature trends. Figure 9 shows near surface air temperature difference between 2000-2014 and 1985-1999 according to ERA Interim reanalysis and model mean data.  Statistical significance for model data was estimated using t-test, 99% confidence

level was used. Reanalysis data look noisier than model mean, but some observed features are reproduced well by the model ensemble. Maximum warming up to 2.5 K appears in Arctic, in Barentz and Kara seas; warming over high and midlatitudes in Eurasia and North America is about 1 K, and lowest warming (or even cooling in reanalysis data) is located in the Southern ocean. Model warming is robust everywhere except some areas in Southern ocean, zone of deep convection in North Atlantic and Zones of Gulf Stream and Kuroshio separation from the shore, where natural variability is high. In the Pacific, the observed pattern connected with the PDO, is not reproduced in model mean data as well as in any individual model run. Figure 10 represents the near surface temperature model trend in two experiments (blue and green) having the maximum and minimum Arctic warming. In the second one (green) there is no Arctic warming at all and even some cooling, and warming over Eurasian and North American midlatitudes also is much smaller than in model average data. Otherwise in the first case (blue) the Arctic warming in some areas is as large as 7 K, and midlatitudinal warming over Eurasia and North America is higher than in model average data.

## 4. Conclusions

Seven historical runs for 1850-2014 with the climate model INM-CM5 were analyzed. It is shown that the magnitude of the GMST rise in model runs agrees with the estimate based on the observations. All model runs reproduce stabilization of GMST in 1950-1970, fast warming in 1980-2000 and a second GMST stabilization in 2000-2014 suggesting that the major factor for predicting GMST evolution is the external forcing rather than system internal variability. Numerical experiments with the previous model version (INMCM4) for CMIP5 showed unrealistic gradual warming in 1950-2014. The difference between the two model results could be explained by more accurate modeling of stratospheric volcanic and tropospheric anthropogenic aerosol radiation effect (stabilization in 1950-1970) due to the new aerosol block in INM-CM5 and more accurate prescription of the TSI scenario (stabilization in 2000-2014) in CMIP6 protocol. Four of seven INM-CM5 model runs simulate acceleration of warming in 1920-1940 in a correct way, other three produce it earlier or later than in reality. This indicates that for the year warming of 1920-1940 the climate system natural variability plays significant role.

No model trajectory reproduces correct time behavior of AMO and PDO indices. Taking into account our results on the GMST modeling one can conclude that anthropogenic forcing does not produce any significant impact on the dynamics of AMO and PDO indices, at least for the INM-CM5 model. In turns, correct prediction of the GMST changes in the 1980-2014 and

increase of ocean heat uptake in 1995-2014 does not require correct phases of the AMO and PDO as all model runs have correct values of the GMST while in at least three model experiments the phases of the AMO and PDO are opposite to the observed ones in that time. The variance explained by PDO and AMO is similar in the model and in the observations. The North Atlantic SST time series produced by the model correlates better with the observations in 1980-2014. Three out of seven trajectories have strongly positive North Atlantic SST anomaly as in the observations (in the other four cases we see near-to-zero changes for this quantity).

The INM-CM5 has the same skill for prediction of the Arctic sea ice extent in 2000-2014 as CMIP5 models including INMCM4. It underestimates the rate of sea ice loss by a factor between the two and three. In one extreme case the magnitude of this decrease is as large as in the observations while in the other the sea ice extent does not change compared to the preindustrial ages. In part this could be explained by the strong internal variability of the Arctic sea ice but obviously the new version of INMCM model and new CMIP6 forcing protocol does not improve prediction of the Arctic sea ice extent response to anthropogenic forcing.

The Model reproduces several observed geographic features of near the surface air temperature trend during the last decades, including Arctic amplification with maximum over Barentz and Kara seas, warming of about 1K over Eurasian and North American midlatitudes and weakest warming over Southern ocean. Case to case variability is very important here as well.

The decrease of stratospheric temperature at 5 hPa during the period of 1979-2014 is successfully reproduced by the model in all experiments. The magnitude of the temperature drop is close to the one for ERA Interim data (2.5 and 3K).

**Acknowledgements**

The study was performed at the Institute of Numerical Mathematics of the Russian Academy of Sciences and supported by the Russian Science Foundation, grant 14-27-00126. Climate model runs were produced at the supercomputer of the Joint Supercomputer Center of the Russian Academy of Sciences and supercomputer Lomonosov at Moscow State University.

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

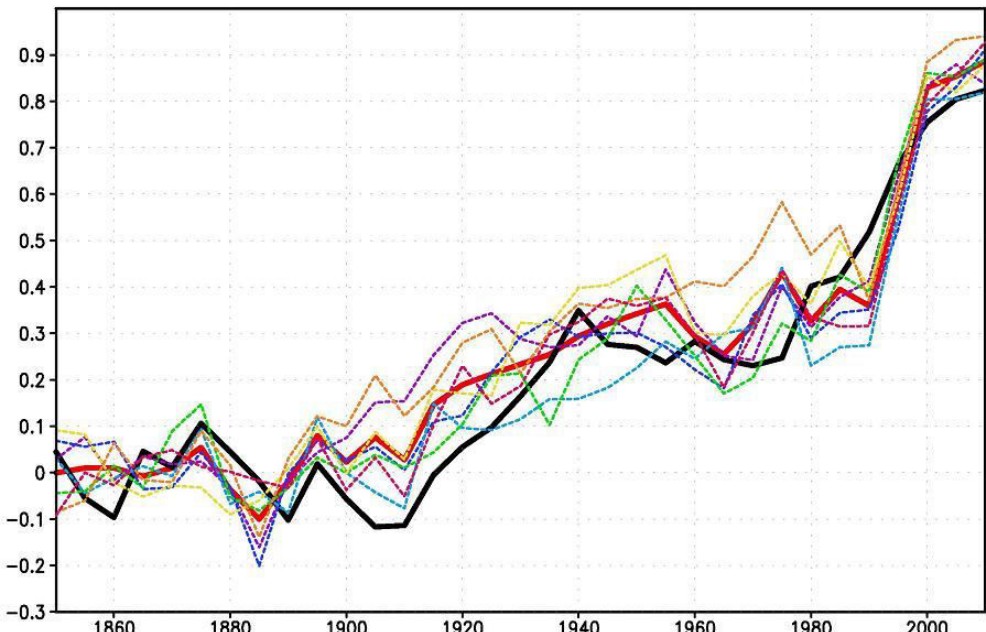

Fig.1. Five year mean GMST (K) anomaly with respect to 1850-1899 for HadCRUTv4 (thick
full black), model mean (thick full red). Dashed thin lines represent data of individual model
runs: 1- purple, 2- dark blue, 3- blue, 4 – green, 5 – yellow, 6 – orange, 7 – magneta.  In this and
next figures numbers at time axis mean first year of five year mean.

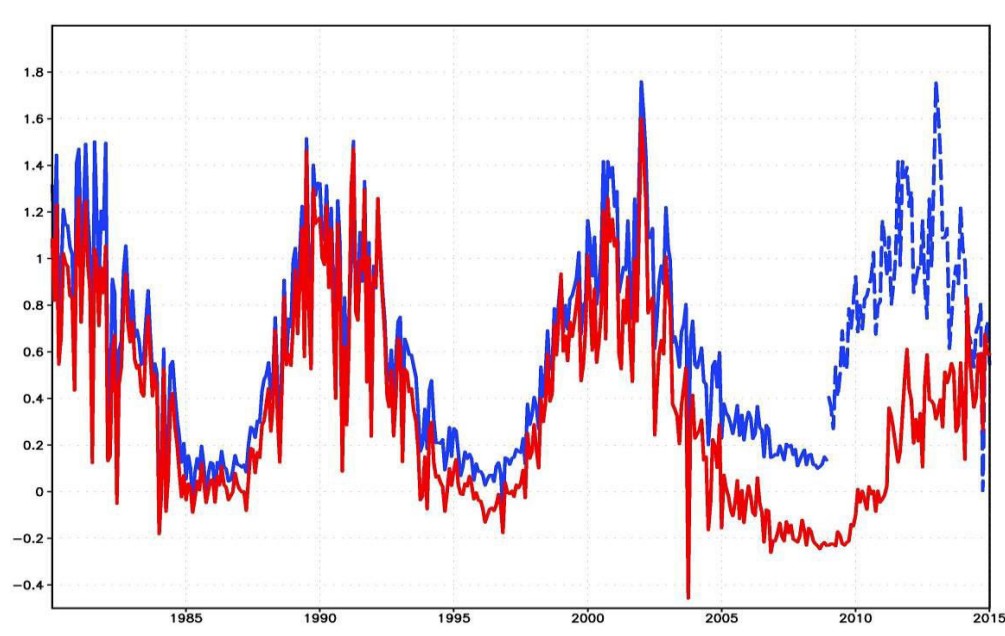

4   Fig.2. Monthly mean TSI anomaly (W/m$^2$) with respect to 1882-1931 recommended for CMIP5
5   (blue; dashed line after year 2008 is the repetition of the data for 1998-2008) and for CMIP6
6   (red).

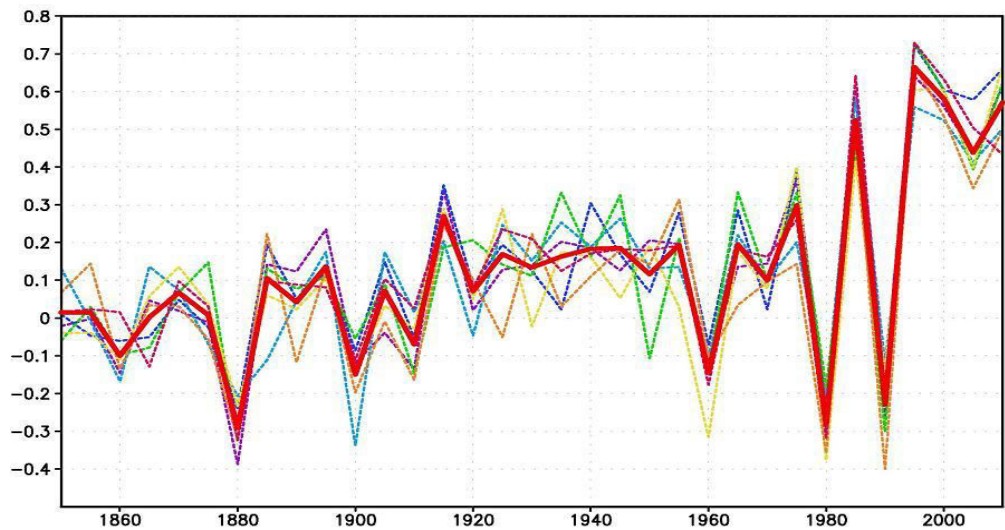

1   Fig.3. Five year mean surface heat flux, W/ m$^2$ (positive downward). Thick full red line

2   represents model mean, dashed thin lines represent data of individual model runs: 1- purple, 2-

3   dark blue, 3- blue, 4 – green, 5 – yellow, 6 – orange, 7 – magneta.

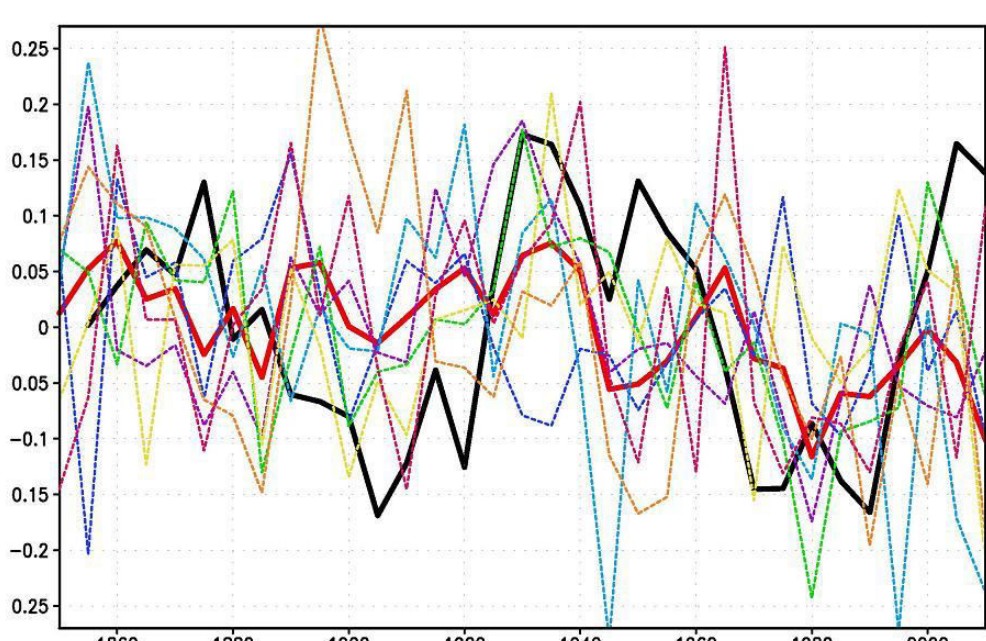

8   Fig.4. Five year mean AMO index (K) for ERSSTv4 data (thick full black), model mean (thick

9   full red). Dashed thin lines represent data of individual model runs. Colors correspond individual

10  runs as in Fig.1

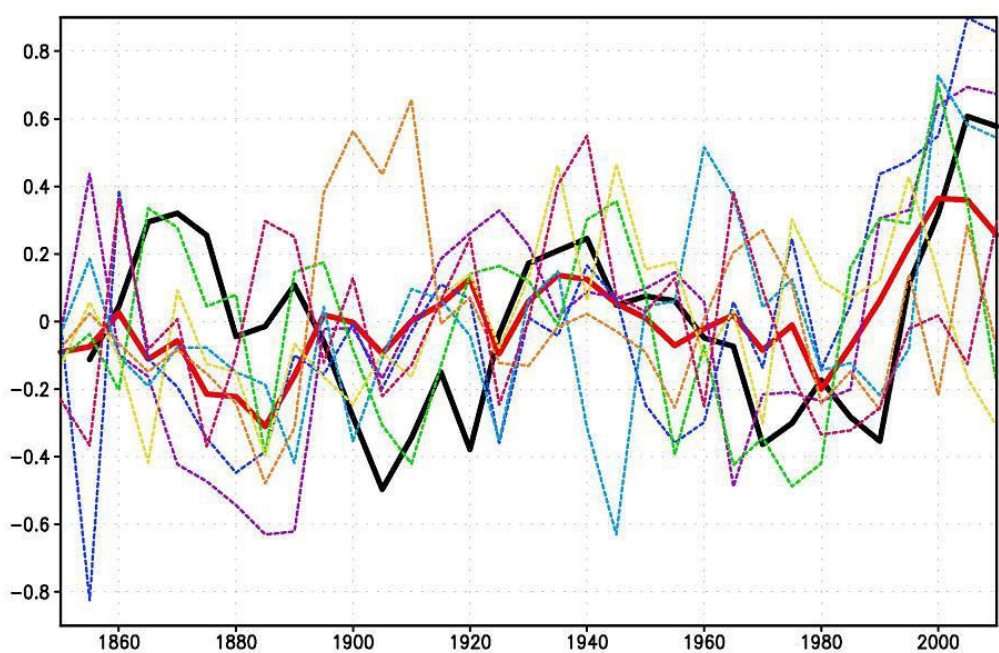

2 Fig.5. Five year mean SST anomaly (K) with respect to 1850-1899 in North Atlantic (45N-65N)

3 for ERSSTv4 data (thick full black), model mean (thick full red). Dashed thin lines represent

4 data of individual model runs. Colors correspond individual runs as in Fig.1

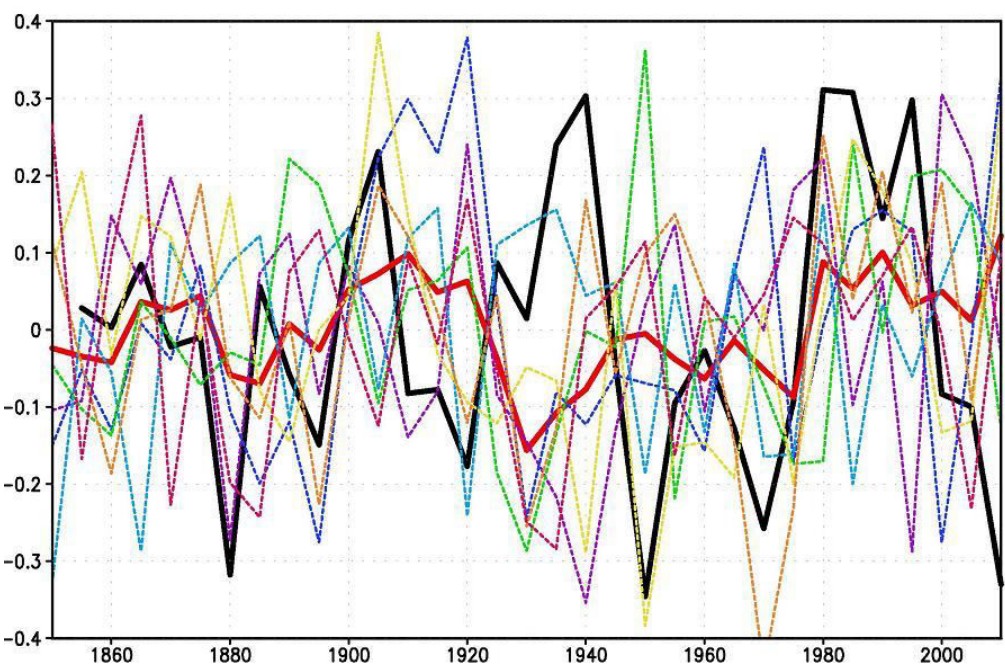

2 Fig.6. Five year mean PDO index (K) for ERSSTv4 data (thick full black), model mean (thick

3 full red). Dashed thin lines represent data of individual model runs. Colors correspond to

4 individual runs as in Fig.1.

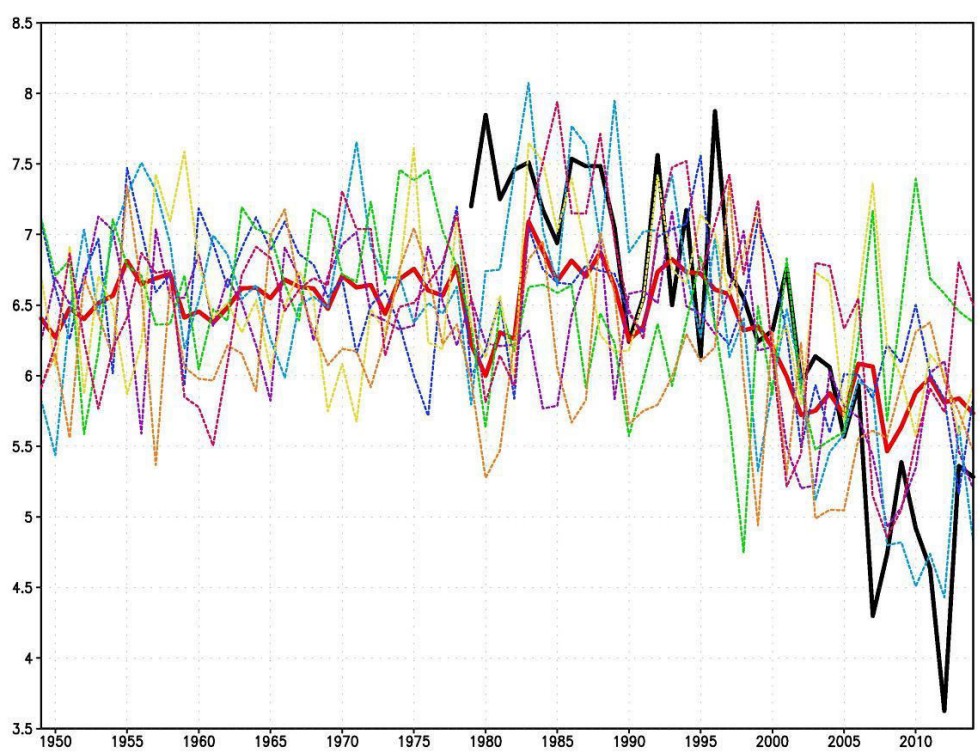

Fig.7. September Arctic sea ice extent ($10^{12}$ m$^2$) for observations (Comiso and Nishio 2008) (thick full black), model mean (thick full red). Dashed thin lines represent data of individual model runs. Colors correspond individual runs as in Fig.1.

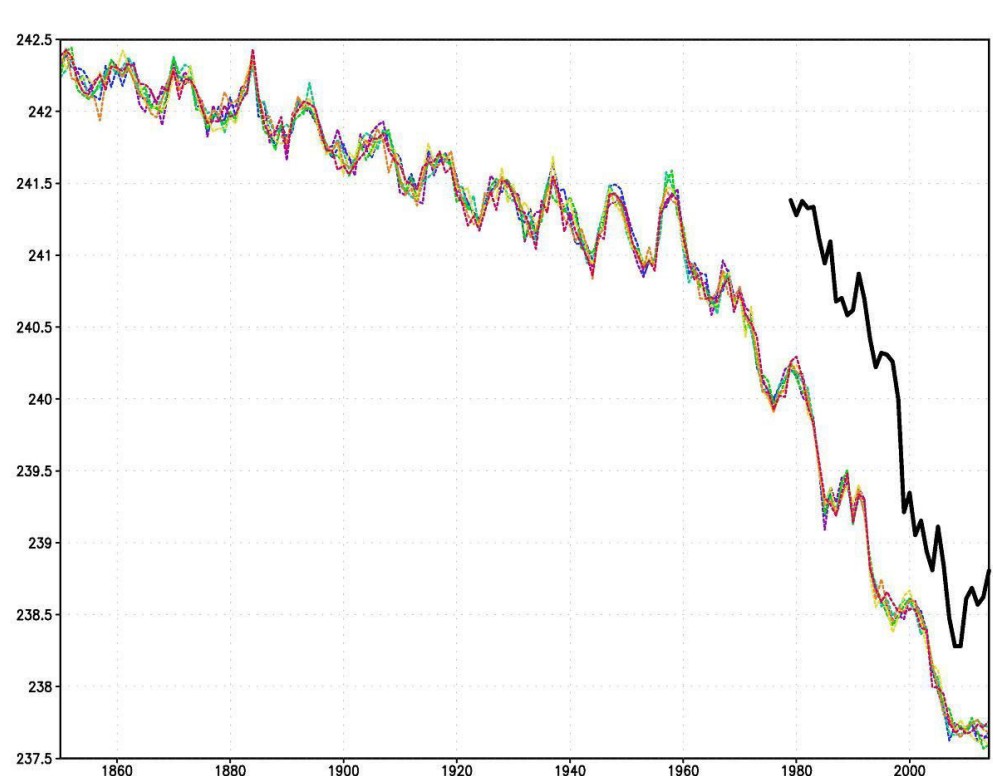

3 Fig.8. Annual mean global mean temperature (K) at 5 hPa for ERA Interim data (black) and

4 model data (dashed color lines).

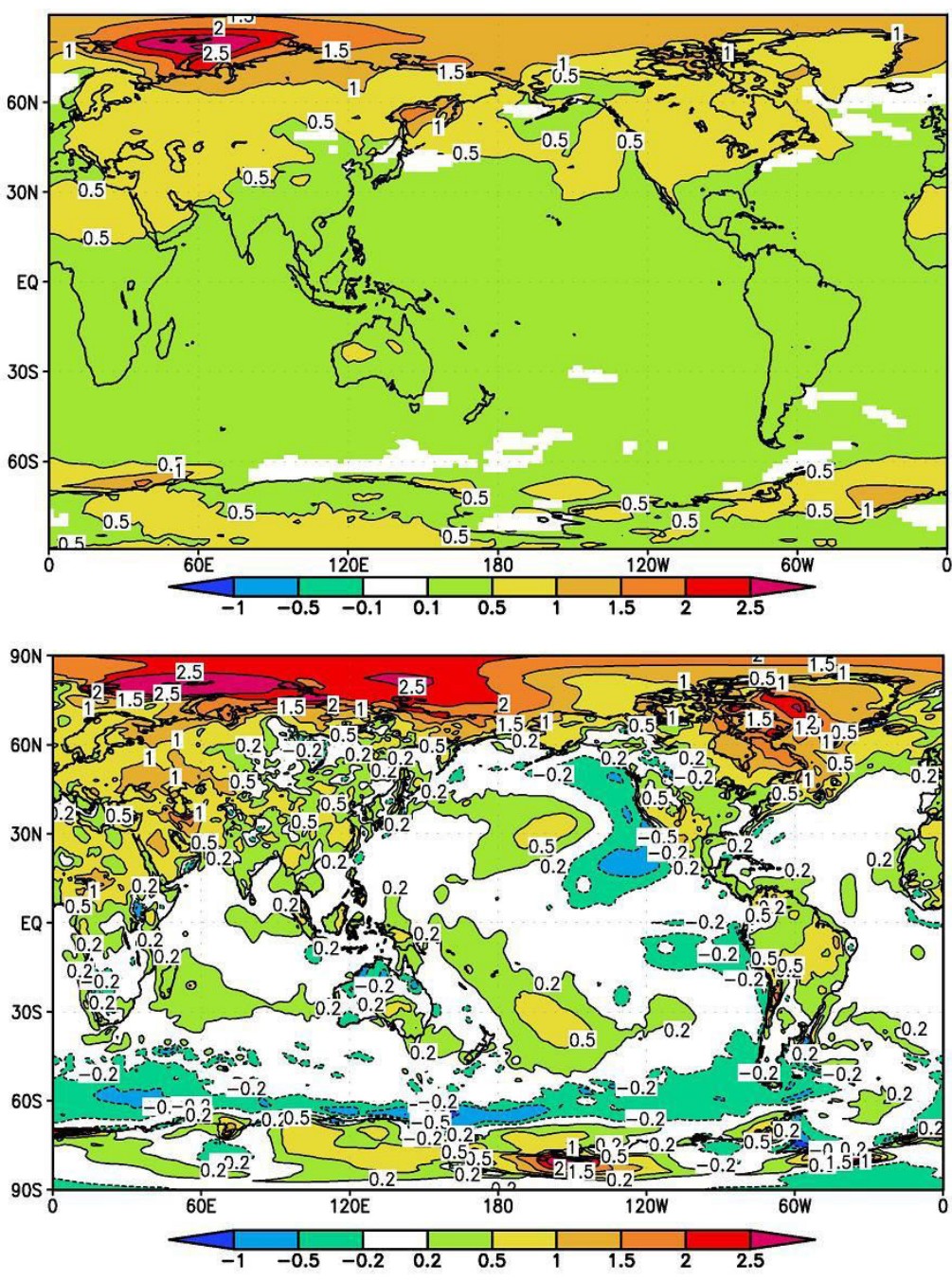

2  Fig.9. Annual mean near surface air temperature (K) in 2000-2014 minus 1985-1999 for model

3  mean data (top), shading represents 99% level of significance, and ERA Interim data (bottom).

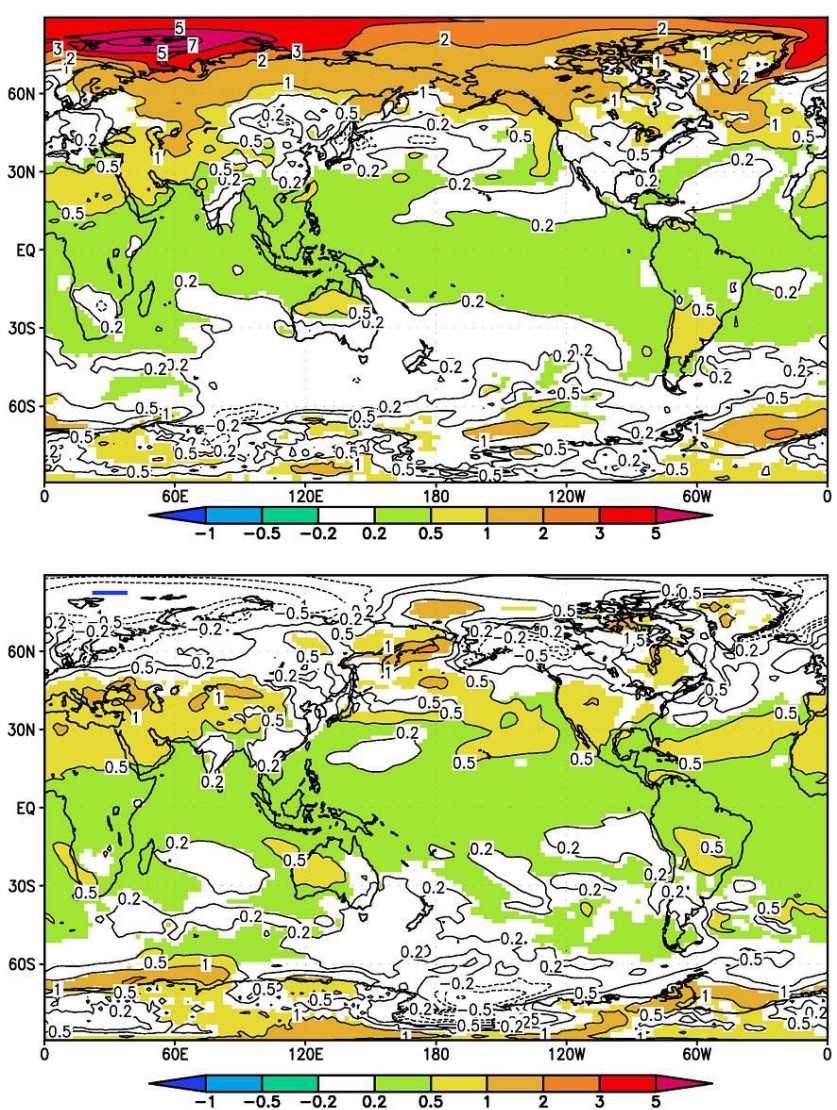

2   Fig.10. Annual mean near surface air temperature (K) in 2000-2014 minus 1985-1999 for model

3   run with highest (top) and lowest (bottom) warming in Arctic. Shading represents 99% level of

4   significance.