# Peer review of "Simulation of observed climate changes in 1850-2014 with climate model 1 2 INM-CM5 3 Evgeny Volodin1\*, Andrey Gritsun1 4 5 1 Institute for Numerical Mathematics, (INM RAS, Gubkina 8, Moscow 119333, Russia) 6 7 \* Correspondence to EvgenyVolodin, Institute for"

_Earth System Dynamics, 2018_

## Referee Comment (RC1) · Anonymous Referee #1 · 15 Jul 2018

The paper presents the results of seven historical experiments with the observed external forcings on the climate system for the period 1850-2014, which were performed with the climate model INM-CM5 within the framework of the international project CMIP6. The main attention is paid to the simulation of the observed evolution of the global mean surface temperature (GMST). Also discussed are the changes in the sea ice cover in the Arctic, the temperature decrease in the stratosphere, and the spatial structure of the surface temperature trend in the last 30 years. The INM climate model has proved itself in the previous stages of the CMIP project, and the results of simulations with the new version of the model (INM-CM5) are of undoubted interest.

The authors use the term "Solar Constant", which is the amount of total solar energy received by unit time and unit area at the mean sun-earth distance. This term is some-

what misleading, because the irradiance is varying in time and hence the majority of contemporary papers (see Kopp and Lean (2011), Kopp (2016), etc.) use "Total Solar Irradiance" or TSI. I would recommend the authors to do the same.

One of the important results of the study is a good ensemble mean reproduction of the observed slowdown in global warming in 2000-2014. The authors attribute this result to more accurate description of the TSI variability in the CMIP6 protocol. It should be noted that this conclusion should be treated with caution. The slowdown in warming begins in 2000, when the discrepancy in the TSI values between the CMIP5 and CMIP6 protocols is still very small (Fig. 2). It is difficult to expect that such a small value can have a significant effect on the change in GMST. Moreover, there is a general agreement in the literature (see Yan et al. (2016) and references in it) that the slowdown of GMST increase in 1998-2013 was a result of the increased uptake of heat energy by the global ocean during these years. Although the slowdown in warming in the model simulations, are obtained when the AMO and PDO indices are incorrectly reproduced in the experiments, it does not indicate that the redistribution of heat in the global ocean could not be the main cause of the slowdown in global warming.

Kopp, G., and J.L. Lean. Geophys. Res. Lett., 38, L01706, 2011, DOI: 10.1029/2010GL045777. Kopp G. J. Space Weather Space Clim., 6, A30, 2016, DOI: 10.1051/swsc/2016025. Yan, X.-H. et al., 2016, Earth's Future, 4, 472-482, doi:10.1002/2016EF000417.

The article clearly lacks punctuation marks, for example in the expression "et al." in references to literature.

---

## Referee Comment (RC2) · Anonymous Referee #2 · 22 Jul 2018

Page 1: lines 23-24 'just in' should be 'in just' line 24 'Spatial' should be 'The spatial' line 25 'of model mean' should be 'of the model mean' and 'close the' should be 'close to the' line 26 'Model ' should be 'The Model' line 32 'A noticeable' should be 'Noticeable' and 'Main' should be 'The main' line 34-34 'reason is anthropogenic increase of' should be 'reason is the increase of anthropogenic' Page 2: line 16 'and North' should be 'and the North' line 17 'produce significant' should be ' produce a significant' line 26 'participated' should be 'that participated' page 3: line 3 ' But' should be 'However' line 26 'for historical' should be 'for the historical' line 29 'so upper' should be 'so the upper' page 4: line 3 't-test' should be 'a t-test' and 'Variance' should be 'The variance' line 9 'Stratospheric' should be 'The stratospheric' Page 5: line 1 'in more' should be 'in a more' line 9 ' Model' should be 'The model' Page 6: line 15 'then' should be

'than' line 16 'the stratospheric' should be 'stratospheric' line 32 'Southern' should be 'the Southern' line 34 'In Pacific, observed' should be 'In the Pacific, the observed' and 'PDO' should be 'the PDO' line 35 'near' should be 'the near' Page 7 line 11 'that magnitude' should be 'that the maginitude' line 19 'of Solar' should be 'of the Solar' line 32 'as the observations' should be ' as in the observations' With regard to the conclusions detailed between lines 24-33 it would be reassuring to note that the percent of variance explained by the PDO and AMO is similar i the model and nature, if that is so. Page 8: line 6 'Model' should be 'The model'

---

## Author Comment (AC1) · 27 Jul 2018

We thank the referee for the careful reading and important comments and we will change the paper accordingly.

1. Referee comment: The authors use the term "Solar Constant", which is the amount of total solar energy received by unit time and unit area at the mean sun-earth distance. This term is some- what misleading, because the irradiance is varying in time and hence the majority of contemporary papers (see Kopp and Lean (2011), Kopp (2016), etc.) use "Total Solar Irradiance" or TSI. I would recommend the authors to do the same.

Author's response: We agree with the reviewer

[Figure]

Author's changes in the manuscript: "Solar constant" will be replaced by "Total Solar Irradiance", or "TSI".

2. Referee comment: One of the important results of the study is a good ensemble mean reproduction of the observed slowdown in global warming in 2000-2014. The authors attribute this result to more accurate description of the TSI variability in the CMIP6 protocol. It should be noted that this conclusion should be treated with caution. The slowdown in warming begins in 2000, when the discrepancy in the TSI values between the CMIP5 and CMIP6 protocols is still very small (Fig. 2). It is difficult to expect that such a small value can have a significant effect on the change in GMST. Moreover, there is a general agreement in the literature (see Yan et al. (2016) and references in it) that the slowdown of GMST increase in 1998-2013 was a result of the increased uptake of heat energy by the global ocean during these years. Although the slowdown in warming in the model simulations, are obtained when the AMO and PDO indices are incorrectly reproduced in the experiments, it does not indicate that the redistribution of heat in the global ocean could not be the main cause of the slowdown in global warming.

Author's response: Authors agree with the reviewer on this (very important) statement. Yes, the ocean heat uptake can play significant role in global warming decrease in 2000-2014 obtained in the model, and, yes, it could happen even if the AMO and PDO have incorrect phases. We will further investigate this question.

Author's changes in the manuscript: Last paragraph at page 4 where the role of TSI is discussed will be rewritten in such a way that TSI decrease after year 2000 is one of possible mechanisms responsible for global warming slowdown. Ocean heat uptake in model experiments will be calculated, and conclusion about its role in global warming slowdown obtained in the model will be added to the paper.

3. Referee comment: The article clearly lacks punctuation marks, for example in the expression "et al." in references to literature.

Author's response: Authors agree with the comment.

Author's changes in the manuscript: Punctuation marks will be checked and corrected.
* * *

---

## Author Response (AR1)

**Response to comment of referee #1**

We thank the referee for the careful reading and important comments and we will change the paper accordingly.

**1. Referee comment**:
The authors use the term "Solar Constant", which is the amount of total solar energy received by unit time and unit area at the mean sun-earth distance. This term is some-what misleading, because the irradiance is varying in time and hence the majority of contemporary papers (see Kopp and Lean (2011), Kopp (2016), etc.) use "Total Solar Irradiance" or TSI. I would recommend the authors to do the same.

**Author's response:**
We agree with the reviewer

**Author's changes in the manuscript:**
"Solar constant" was replaced by "Total Solar Irradiance", or "TSI" everywhere in the text.

**2. Referee comment**:
One of the important results of the study is a good ensemble mean reproduction of the observed slowdown in global warming in 2000-2014. The authors attribute this result to more accurate description of the TSI variability in the CMIP6 protocol. It should be noted that this conclusion should be treated with caution. The slowdown in warming begins in 2000, when the discrepancy in the TSI values between the CMIP5 and CMIP6 protocols is still very small (Fig. 2). It is difficult to expect that such a small value can have a significant effect on the change in GMST. Moreover, there is a general agreement in the literature (see Yan et al. (2016) and references in it) that the slowdown of GMST increase in 1998-2013 was a result of the increased uptake of heat energy by the global ocean during these years. Although the slowdown in warming in the model simulations, are obtained when the AMO and PDO indices are incorrectly reproduced in the experiments, it does not indicate that the redistribution of heat in the global ocean could not be the main cause of the slowdown in global warming.

**Author's response:**
Authors agree with the reviewer on this statement. Yes, the ocean heat uptake can play significant role in global warming decrease in 2000-2014 obtained in the model, and, yes, it could happen even if the AMO and PDO have incorrect phases. We investigated this question and found that it can be really the case.

**Author's changes in the manuscript:**
In abstract (p.1, line 20) we mentioned about this.
In p.4, lines 1-7 we added discussion about this and presented Fig.3 with heat flux to ocean that shows ocean heat uptake in 1995-2010 higher than in any other time.
In summary (p.8, line 3) we mention about the importance of ocean heat uptake.

**3. Referee comment**:
The article clearly lacks punctuation marks, for example in the expression "et al." in references to literature.

**Author's response:**
Authors agree with the comment.

**Author's changes in the manuscript:**
Punctuation marks and in particular et al. were cheched.

**Response to comment of referee #2**

We thank the referee for the careful reading and important comments and we will change our paper accordingly.

**1. Referee comment**:
Page 1: lines 23-24 'just in' should be 'in just' line 24 'Spatial' should be 'The spatial' line 25 'of model mean' should be 'of the model mean' and 'close the' should be 'close to the' line 26 'Model ' should be 'The Model' line 32 'A noticeable' should be 'Noticeable' and 'Main' should be 'The main' line 34-34 'reason is anthropogenic increase of' should be 'reason is the increase of anthropogenic' Page 2: line 16 'and North' should be 'and the North' line 17 'produce significant' should be ' produce a significant' line 26 'participated' should be 'that participated' page 3: line 3 ' But' should be 'However' line 26 'for historical' should be 'for the historical' line 29 'so upper' should be 'so the upper' page 4: line 3 't-test' should be 'a t-test' and 'Variance' should be 'The variance' line 9 'Stratospheric' should be 'The stratospheric' Page 5: line 1 'in more' should be 'in a more' line 9 ' Model' should be 'The model' Page 6: line 15 'then' should be 'than' line 16 'the stratospheric' should be 'stratospheric' line 32 'Southern' should be 'the Southern' line 34 'In Pacific, observed' should be 'In the Pacific, the observed' and 'PDO' should be 'the PDO' line 35 'near' should be 'the near' Page 7 line 11 'that magnitude' should be 'that the maginitude' line 19 'of Solar' should be 'of the Solar' line 32 'as the observations' should be ' as in the observations' With regard to the conclusions detailed between lines 24-33 it would be reassuring to note that the percent of variance explained by the PDO and AMO is similar i the model and nature, if that is so. Page 8: line 6 'Model' should be 'The model'

**Author's response:**
We agree with the reviewer

**Author's changes in the manuscript:**
All proposed changes are introduced  in the text and are marked.

[revised manuscript text omitted]